# Effect of the Marine Exercise Retreat Program on Thyroid-Related Hormones in Middle-Aged Euthyroid Women

**DOI:** 10.3390/ijerph20021542

**Published:** 2023-01-14

**Authors:** Hangjin Byeon, Yesol Moon, Seoeun Lee, Gwang-Ic Son, Eunil Lee

**Affiliations:** 1Department of Public Health, College of Medicine, Korea University, Seoul 02841, Republic of Korea; 2Department of Biomedical Science, Graduate School, Korea University, Seoul 02841, Republic of Korea; 3Department of Preventive Medicine, College of Medicine, Korea University, Seoul 02841, Republic of Korea

**Keywords:** thyroid hormones, TSH, Free T4, marine retreat program, blue-space, heart rate variability, euthyroid

## Abstract

This study aimed to investigate the effects of a marine exercise retreat program on thyroid-related hormone levels. A total of 62 middle-aged euthyroid women participated in a 6-day marine exercise retreat program. Using thyroid-stimulating hormone (TSH) and free thyroxine (fT4) hormone levels, the participants were divided into high and low-hormone-level groups. Despite decreased TSH and fT4 levels after the program, the factors influencing changes in each group were different. TSH levels were influenced by changes in the normalized low frequency (nLF) of heart rate variability and carbon monoxide (CO) from all the participants, and changes in body fat percentage, nLF, and nitrogen dioxide (NO_2_) exposure level in the high TSH group. fT4 levels were influenced by changes in body mass index (BMI), NO_2_ exposure, and particulate matter diameter of 10 µm or less (PM_10_) exposure in all participants. Changes in BMI and CO exposure influenced the low fT4 group. Lastly, changes in the exercise stress test affected the high fT4 group. Thus, the marine exercise retreat program affected euthyroid thyroid-related hormone levels, and influencing factors differ depending on the initial value of the hormone.

## 1. Introduction

Thyroid function plays a pivotal role in maintaining the human physiological system related to metabolism, growth, development, and homeostasis [1]. Thyroid hormones are released from the thyroid glands and maintained in a certain amount in the bloodstream through a negative feedback loop mechanism [2]. Abnormal hormone secretion from the thyroid gland leads to hyperthyroidism or hypothyroidism in severe or subclinical forms [3]. Hypothyroidism may increase the risk of cardiovascular disease and all-cause mortality [4]. Similarly, some studies have shown that hyperthyroidism increases the risk of cardiovascular disease and cancer [5]. Not only do thyroid-related diseases increase the risk of mortality, but hormone levels within the normal range are also associated with mortality [6].

Diagnosing thyroid dysfunction is made by referring to the upper and lower limits of hormones such as thyroid-stimulating hormone (TSH) and free thyroxine (fT4). However, several studies have shown that euthyroid hormone levels are associated with various health risks. Ruhla et al. [7] reported that patients with TSH levels in the upper normal range had an increased risk of metabolic syndrome. Kim et al. [8] also reported that high TSH levels within the normal range may contribute to initiating thyroid carcinogenesis. Additionally, a high TSH level within the normal range affects both physical and psychological health. A previous study reported that a high-normal TSH concentration was associated with low cognitive function [9]. Additionally, Chueire et al. [10] reported that patients with elevated TSH serum levels, including those with normal TSH levels, may be closely related to mood disturbances and depression in older adults. Due to the fact that the high health risks associated with TSH levels, many studies argue that TSH levels must be carefully managed and that the normal range of TSH must be reassigned to a lower and narrower range than that from current standards [11,12].

Air pollutants cause thyroid diseases and affect thyroid hormone levels. Air pollutants are mainly generated by industrial activities, fuel combustion in automobiles and power plants, and construction. As urbanization progresses, the degree of pollution becomes more severe with increasing concentration in places where urbanization has progressed further. Park et al. [13] reported that air pollutants correlate with thyroid cancer. Zhang et al. [14] also reported that air pollutant levels might increase the risk of thyroid nodules in a study of 4.9 million Chinese adults. Air pollutants not only cause thyroid disease but also act as endocrine disruptors and interfere with hormone secretion. Living in areas with high levels of air pollutants, such as nitrogen dioxide (NO_2_), carbon monoxide (CO), and particulate matter with an aerodynamic diameter of 10 µm or less (PM_10_), is strongly associated with thyroid function and hormone release [15]. Moreover, Irizar et al. [16] reported that exposing pregnant women to air pollutants such as PM_2.5_ was positively associated with infant thyroxine levels at birth. This would lead to thyroid dysfunction.

The sedentary lifestyle of modern society causes obesity and lack of exercise, which interferes with thyroid hormone secretion [17]. A sedentary lifestyle restricts physical movement and is strongly associated with obesity. Obesity and body mass index (BMI) are major factors disturbing thyroid function [18]. A sedentary lifestyle not only affects obesity but also reduces physical activity, ultimately leading to a decrease in physical ability. Some studies have reported that people diagnosed with thyroid-related diseases are relatively physically inactive [19]. However, a high physical performance due to regular physical activity was found to be beneficial for thyroid function [20].

Psychological stress would have an impact on thyroid function. As the number of people residing in urbanized places increases, more people become vulnerable to stress. Urbanized places are full of physical stressors, such as noise, heat, and toxic substances, and mental stressors from work and social relationships. It is well known that this stress induces cortisol release in the human body from the hypothalamus-pituitary-adrenocortical (HPA) system. Some studies have shown that thyroid hormone secretion is suppressed due to increased cortisol [21]. Because of this stress, thyroid function is downregulated, affecting thyroid hormone concentration [22]. Additionally, anxiety from stressful situations may increase TSH levels and disrupt hormonal regulation [23]. Under chronic stress, the sympathetic nervous system (SNS) is constantly activated without counteracting by the parasympathetic nervous system (PNS) [24]. The autonomic nervous system (ANS) actively interacts with thyroid hormones and shows imbalanced nervous function in patients with hypothyroidism [25]. Heart rate variability (HRV) analysis is a widely applied method for assessing psychological stress and sympathovagal balance [26]. HRV is also an ANS function that consists of the SNS and PNS. Many studies have evaluated the status of the SNS and PNS of the ANS using HRV and found a high correlation with thyroid-related diseases such as hypo- and hyperthyroidism [27,28,29].

Recently, some studies have accumulated evidence on the health effects of spending time and performing exercise and activities in natural environments, such as forest bathing, forest treatment, and marine therapy [30,31]. Additionally, avoiding an urbanized environment has numerous positive and restorative physiological and psychological benefits [32]. Hartig et al. [33] reviewed some studies regarding the effect of spending time in the natural environment on health and presented major beneficial pathways, such as better air quality, physical activity, social cohesion, and stress reduction. Moreover, exercise and activities in the natural environment exert beneficial effects on human immune function and cardiovascular conditions [34,35]. Aside from the physiological benefits, there are more reports on the psychological benefits of the natural environment. Numerous studies have reported that forest activities help reduce mental stress and anxiety [36,37].

There has been increasing interest in research on the potential health benefits of the marine environment in recent years. The marine environment contains a wealth of beneficial elements, including iodine and selenium, which are essential for human health, and exhibits a high quality of air [38,39,40,41]. Furthermore, ultraviolet rays from the marine environment stimulate the production of vitamin D, which helps to strengthen the immune system [42]. In addition, activities in the marine environment, such as walking on the sandy beach, have been shown to have excellent effects on stress reduction and mental health [43,44]. However, there have been few studies on thyroid function improvement using programs in the natural environment, especially in the marine environment. In fact, no study has investigated the effects of activity in the marine environment on thyroid hormones. Moreover, most studies on thyroid-related hormones have focused on people who have already been diagnosed with the disease. However, as there are health risks associated with hormone levels, studies on thyroid-related hormone changes in the normal range group are also important. We investigated the effects of a marine exercise retreat program on thyroid function by observing thyroid-related hormones, such as TSH and fT4. We divided the participants into two groups according to their TSH and fT4 levels and analyzed the association of hormone level changes with physical changes, such as obesity and exercise stress, psychological changes, such as stress and ANS, and environmental change, such as air pollutant level. We also investigated the factors affecting changes in thyroid-related hormone levels after the marine exercise retreat program.

## 2. Materials and Methods

### 2.1. Participants

In this study, we recruited middle-aged euthyroid women to participate in a marine exercise retreat program. We excluded participants such as current smokers, those having problems with moderate physical activity, and those with a thyroid-related surgery and treatment history. A total of 72 people participated, and 62 with normal TSH and fT4 levels were selected as the study participants. The participants were middle-aged adult women between 49 and 71 years old. Participants were recruited according to three residential classifications (urban, suburban, and rural areas). Participants residing in Seoul city were selected for the urban area, and those living in Gyeonggi-do and Gwangju were selected for the suburban areas of Korea. For rural areas, participants were recruited from Wando, Korea. Information on basic disease history, including hypertension, diabetes, hyperlipidemia, and metabolic syndrome, was collected. The purpose and procedure of the experiment were fully explained in both spoken and written forms. The study was approved by the institutional review board (IRB) of Korea university (KUIRB-2021-0094-06). Detailed objectives and procedures of the study were informed, and consent was obtained from the participants.

### 2.2. Marine Exercise Retreat Program

The program was conducted on Sinji island in Wando-gun, located in the southern sea of Korea. Most of the programs were held at Sinji Myeongsasimni beach. The beach has wide silver-white sand of 3800 m in length and 150 m in width, with gentle and wide slopes. The main program lasted for 5 d. Participants were divided into four groups and underwent four sessions. The arrival on the site a day before the program when participants began to acquire basic information was considered Day 0. In the program, all participants underwent Nordic walking for 2 h in the morning (9:30–11:30) and a beach healing program for 2 h in the afternoon (15:30–17:30) every day. The beach healing program consisted of meditation at the beach, light exercise, and stretching. Daily meals provided to the participants were prepared based on general nutritional standards, mainly using seafood produced on Wando-gun, designed by a nutritionist from the Department of Food and Nutrition of Inha University. Participants were also encouraged to go to bed and wake up at the same time each day.

### 2.3. Measurements

#### 2.3.1. Physical Body Measurements and Exercise Stress Score

Weight, BMI, and body fat percentage were measured using an Inbody-270 Body Composition Analyzer (Inbody, Seoul, Republic of Korea) before and after the program. A stadiometer was used to measure height. All participants’ waist and hip circumferences were measured, and the waist-to-hip ratio (WHR) was calculated. Blood pressure was measured using a HEM-7121 (Omron, Kyoto, Japan) blood pressure monitor before and after the program.

We also measured the exercise stress scores of the participants before and after the program. The exercise stress score was calculated using exercise intensity in the Karvonen scale. The chair test was used to measure exercise stress. The participants repeatedly sat and stood up from the chair 100 times for 5 min, and the maximum heart rate (HR) during the test was recorded. According to the equation below, the lower the score, the better the physical ability.
(1)$$\mathrm{Exercise}\;\mathrm{stress}\;\mathrm{score}=\frac{\left(\mathrm{Maximum}\;\mathrm{heart}\;\mathrm{rate}\;\left[\mathrm{bpm}\right]-\mathrm{Resting}\;\mathrm{heart}\;\mathrm{rate}\;\left[\mathrm{bpm}\right]\right)}{\left(\left[220-\mathrm{Age}\right]-\mathrm{Resting}\;\mathrm{heart}\;\mathrm{rate}\right)\times 100}$$

#### 2.3.2. Thyroid-Related Hormones

Thyroid function was assessed by measuring serum levels of blood samples. Blood samples were collected from each participant in the fasting state on the morning of the first and last days of the program. Blood samples were sent to the LabGenomics Laboratory (Korea), and serum TSH (mIU/L) and fT4 (ng/dL) concentrations were measured using a chemiluminescence immunoassay (Atellica Solution Immunoassay and Clinical Chemistry Analyzer; Siemens, Munich, Germany).

#### 2.3.3. Heart Rate Variability

Electrocardiography (ECG) was performed on the morning of the first and last days of the program. ECG was measured 30 min after participants woke up from sleep. Participants were kept in a fasting state, and measurement was started after 15 min of stabilization time without moving or talking while sitting comfortably. Silver/silver chloride (Ag/AgCl) electrodes were attached to the participant’s left and right arms for the lead electrodes and the left leg for the ground electrode. The signal was acquired using an AD8232 (analog device) sensor chip sampling at 100 Hz frequency. For the signal processing and HRV parameter calculations, we used an in-house MATLAB script. Typically, HRV is analyzed in the time and frequency domains. For time-domain metrics, we obtained HR, a standard deviation of NN intervals (SDNN), and root mean square of successive NN interval differences (RMSSD). For frequency domain analysis, high-frequency (HF; 0.15–0.40 Hz) power, low-frequency (LF; 0.05–0.15 Hz) power, and the ratio of LF to HF (LF/HF) were used. Normalized LF (nLF) and HF (nHF) were derived from the calculation (nLF = LF/[LF + HF], nHF = HF/[LF + HF]).

### 2.4. Residential Pollutant Exposure

To measure the exposure level of air pollution in the daily life of the participants, information on the average annual air pollution level in each residential area was obtained. Six air pollutants (sulfur dioxide [SO_2_], NO_2_, ozone [O_3_], CO, PM_10_, and PM_2.5_) were collected from the residential areas of the participants. Pollutant concentration was measured using the Korea National Ambient Air Quality Monitoring Information System, and information was obtained from the AirKorea website (https://www.airkorea.or.kr, accessed on 2 September 2021). The pollutant level data was acquired from the closest monitoring site of the participants’ residential areas.

### 2.5. Statistical Analysis

The participants were divided into two groups according to the level of TSH and fT4 before the program started: (1) TSH grouping: low (0.5 < TSH < 2.3 mIU/L, *n* = 31) and high groups (2.3 < TSH < 5.0 mIU/L, *n* = 31); and (2) fT4 grouping: low (0.9 < fT4 < 1.24 ng/dL, *n* = 31) and high groups (1.24 < fT4 < 1.56 ng/dL, *n* = 31). All statistical analyses were conducted in R (version 4.1.1), using R Studio version 1.4.1717. The participants’ baseline characteristics and measurements were presented as means and standard deviation (SD) for continuous variables. Counts, and percentages were used for categorical variables. Normality tests were conducted using the Shapiro–Wilk test. To compare two groups with continuous variables, the Student’s *t*-test or Mann–Whitney test was used as appropriate after the normality test. Pearson’s correlation analysis was used for the correlation analysis. Additionally, for comparison before and after the program in the same group, paired *t*-test and Wilcoxon test were used after normality test. Categorical variables were compared using the chi-squared test or Fisher’s exact test. Multivariate linear regression analysis was performed to investigate the effects of these factors on thyroid hormone levels. Statistical significance was set at *p* < 0.05.

## 3. Results

To investigate differences in characteristics between the low and high groups of each thyroid-related hormone, baseline characteristics, including disease records, physical body measurements, and thyroid-related hormone levels of participants, were analyzed before the program (Table 1). There were no significant differences between the low and high groups of both TSH and fT4 groups except for each group’s TSH and fT4 levels.

Concentrations of six air pollutants (SO_2_, NO_2_, CO, O_3_, PM_10_, and PM_2.5_) were analyzed to identify concentration differences in the participants’ residential areas according to the thyroid hormone group. No pollutants showed significant differences between the low TSH and high TSH groups (Table 2). However, the concentrations of SO_2_, NO_2_, and PM_10_ in the low fT4 group were significantly higher than that of the fT4 high group. O_3_ concentration in the high fT4 group was significantly higher than that in the low fT4 group.

Baseline HRV according to TSH and fT4 levels was analyzed before the start of the program. No significant differences were found between high and low TSH and fT4 groups in any HRV parameters (Table 3). However, normalized LF, normalized HF, and LF/HF ratio parameters significantly correlated with the level of low and high TSH groups (Figure 1). The nLF and LF/HF ratios negatively correlated with TSH levels in the low TSH group. In contrast, nLF and LF/HF ratio showed a positive association with TSH levels in the high TSH group, which showed an opposite trend to that of the low TSH group. However, there was no significant correlation between the total fT4 group and both the low and high groups (Appendix A).

To observe the effect of the marine exercise retreat program, the changes in physical body measurements and levels of thyroid-related hormones of participants according to the subgroup before and after the program were compared (Table 4). No significant change in physical body measurement was observed after the program than that of before the program, except for thyroid-related hormones. Both TSH and fT4 levels in the total study group decreased significantly after the retreat program. However, only the high TSH group showed a significant decrease in TSH value, whereas the low TSH group showed an increase, although the difference was not significant. Both the high and low TSH groups showed a significant reduction in fT4 levels. Changes in the basic characteristics of the participants in the TSH subgroup were observed only in the BMI. Only the BMI in the high TSH group decreased significantly. In the fT4 subgroups, TSH and fT4 levels were significantly decreased in the high fT4 level group, but no significant TSH and fT4 changes were observed in the low fT4 group. We also analyzed changes in each HRV parameter after the marine exercise retreat program (Table 5). Only Mean HR increased after the program for all subgroups of TSH and fT4. No other HRV parameters changed after the program.

Pollution exposure in residential areas before the program was compared with that during the program (Table 6). The concentrations of all six pollutants were significantly different from the average pollutant concentrations in the residential areas of the participants. Pollution at the program site was significantly lower than that in the residential areas, except for O_3_. The SO_2_ concentration of low TSH and high fT4 groups were not significantly different when comparing the before-the-exercise-program and after-the-exercise-program values.

Correlations between changes in TSH (ΔTSH) and fT4 (ΔfT4) levels before and after the program and other characteristics were analyzed. No physical body measurement changes significantly correlated with ΔTSH (Appendix A). The correlation between HRV parameters and TSH changes was also analyzed (Appendix A) and only parameters in the frequency domain showed a significant correlation. There was a positive correlation between nLF (ΔnLF) and ΔTSH in the total participants (*r* = 0.356, *p* < 0.01). Moreover, both the low and high TSH subgroups showed significant correlations between TSH changes and ΔnLF (Figure 2a). The change in LF/HF ratio also positively correlated with ΔTSH (*r* = 0.274, *p* = 0.03). However, only the high group showed a significant positive correlation (Figure 2b).

For ΔfT4 correlation analysis, the low fT4 group showed a positive correlation between changes in body fat % (ΔBody fat %) and ΔfT4 levels. Meanwhile, no correlation was found in the high fT4 group (Appendix A). Changes in diastolic blood pressure (ΔDBP) were negatively associated with ΔfT4 in all the groups. However, no subgroups were correlated with ΔfT4 and ΔDBP. In the high fT4 group, changes in exercise stress (ΔExercise stress) were also negatively correlated with ΔfT4. The correlation between ΔfT4 and HRV parameters was also analyzed (Appendix A). However, no significant correlation was observed between the total and subgroups.

Multiple stepwise regression analysis was conducted to analyze the factors influencing ΔTSH levels through the marine exercise retreat program. In the total TSH group, ΔnLF and CO (ΔCO) significantly influenced ΔTSH (Table 7). Moreover, ΔBody fat%, ΔnLF, and changes in NO_2_ (ΔNO_2_) positively influenced ΔTSH in the high TSH group. Changes in BMI (ΔBMI), ΔNO_2_, and changes in PM_10_ (ΔPM10) were negatively associated with ΔfT4 in the total group (Table 8). Additionally, ΔBMI and ΔCO influenced ΔfT4 in the low fT4 group. In contrast, only ΔExercise stress was negatively associated with ΔfT4 in the high fT4 group.

## 4. Discussion

In this study, we recruited middle-aged women because thyroid-related disease is one of the prevalent diseases in women, and its risk increases with age [30]. Women experience rapid hormonal changes as they age, such as menopause, and the decline in estrogen levels results in hypothyroidism [31]. Some studies have reported that metabolic syndrome significantly increases as TSH levels increase [45,46]. This indicates that even in people diagnosed with normal thyroid function, health risks vary depending on their hormone levels. Hence, we focused on middle-aged euthyroid women as participants in the intervention program.

Although thyroid-related hormones are within the normal range, it poses a different risk to health depending on the level. For example, several studies have found that the risk of metabolic syndrome differs according to TSH levels, and a strong correlation exists [43,47,48]. Additionally, several studies have reported that fT4 is associated with health risks, such as metabolic syndrome [49] and non-thyroidal illness syndrome [50]. As such, even within the normal range, risks and characteristics differ depending on hormone levels; therefore, TSH and fT4 were divided into high and low groups according to the initial TSH and fT4 levels.

We analyzed differences between the high and low TSH and fT4 levels before the program. There were no differences in the baseline characteristics and HRV parameters between the high and low groups. However, correlations between baseline nLF and TSH levels in the high and low groups were observed. The low and high groups exhibited negative and positive correlations, respectively. LF/HF ratio also correlated similarly as that of nLF. As the nLF and LF/HF ratios are indicators of the relative level of sympathetic activity compared with parasympathetic activity, the SNS is activated more than PNS as the TSH level goes to both ends of the euthyroid TSH range. Brusseau et al. [51,52] reported that the nLF and LF/HF ratio increased in hyperthyroid patients with decreased TSH levels, while the nLF and LF/HF ratio increased in hypothyroid patients with increased TSH levels. Although the results of the study did not enlist euthyroid patients, they are consistent with the opposite trend in the nLF and LF/HF ratio in the high and low TSH groups.

After the program, both TSH and fT4 levels significantly decreased. In particular, both TSH and fT4 levels showed a significant decrease in each high group. Our results of TSH reduction after the program were similar to that of previous studies on TSH change after interventions such as exercise and diet. Altaye et al. [53] reported that aerobic exercise had a reduced impact on TSH levels in adolescents with intellectual disabilities. Moreover, Amin et al. [54] reported that interventions, including a dietary program, physical activity, and decreasing sedentary lifestyle, affected TSH reduction in adolescents with obesity. Croce et al. [55] also reported that TSH levels decreased after dietary/lifestyle interventions. However, most studies related to fT4 decrease after our program showed that fT4 increased or did not change as TSH decreased [56,57]. We assume that additional iodine intake may reduce the secretion of fT4 by the Wolff–Chaikoff effect. This is the effect of excessive iodine intake inhibiting iodine absorption into the thyroid and reducing thyroid hormone production [58]. The marine environment of marine retreat programs has different characteristics from those therapies and programs using natural environments, such as forests. The sea breeze and air are typical examples that contain more iodine than that of other regions, which people absorb by respiration [59]. In particular, during the program, the participants maintained a diet consisting mainly of seafood; therefore, they may have consumed more iodine than usual. Since some studies showed that a small increase in dietary iodine may significantly reduce T4 and T3, this may also apply to our results [60]. Our finding that fT4 levels decreased after the program may have been influenced by this phenomenon.

The TSH hormone level is considered the most important thyroid trait to assess thyroid status, since subtle status changes in the thyroid can be captured by measuring TSH levels [61]. This normal TSH range is approximately 0.3–5.0 mIU/L from most references, but many studies recently insist that the optimal or ideal range for TSH should be narrower and lower at approximately 0.4–2.5 mlU/L based on clinical results of thyroid function [62,63]. Additionally, many studies have shown that low-normal TSH serum levels have many beneficial effects on health. Goldman et al. [64] reported that a low-normal level of the current reference range of serum TSH reduces the risk of atrial fibrillation or heart failure. Moreover, Laclaustra et al. [65] reported a lower risk of metabolic syndrome in the lower quintile than that in the higher quintile of the normal TSH range. In this respect, the TSH level of the total group significantly decreased to the optimal range, which has a positive effect on many aspects. Moreover, the reduction in TSH levels in the high TSH group can be interpreted as a positive effect when the optimal TSH range is lower and narrower than that of the normal range.

As participants progressed through the program, they were continuously exposed to ambient air throughout the program. Except for O_3_, all other air pollution levels were lower than those in the residential areas of the participants, indicating that the program was conducted in areas with much better air quality. Air pollutants directly affect thyroid function. In our study, changes in exposure levels of CO and NO_2_ positively affected ΔTSH in the TSH total and high TSH groups, respectively. In addition, exposure level change of CO, NO_2_, and ΔPM_10_ negatively affected ΔfT4. This result is consistent with the report by Kim et al. [15] that CO and NO_2_ positively correlated with TSH and negatively correlated with fT4. Kim et al. analyzed nationwide data across all thyroid hormone ranges for all sexes and age groups. This indicates that this association also applies to individuals in the euthyroid range. Although no studies have found an association between PM10 and fT4 levels, Howe et al. [66] found that prenatal PM_10_ exposure is associated with total thyroxine levels. Moreover, Park et al. [13] reported the association of PM10 exposure with the occurrence of thyroid cancer. Although the mechanism between thyroid-related hormone levels and air pollutant levels has not been clearly elucidated, it is presumed that exposure to low air pollutant concentrations during a marine retreat program might reduce oxidative stress, and the thyroid is affected by reduced toxic effects [67].

The marine exercise retreat program includes physical activity near coastal areas. These physical activities produce various positive effects, such as weight and body fat loss. In our results, ΔBody fat % of the TSH high group was an effective factor for ΔTSH after the program. This result is consistent with that of a previous study showing that weight loss or decreased body fat mass is associated with changes in thyroid hormone levels. Mukherjee et al. [68] investigated the positive association between TSH levels and body fat % in female euthyroid participants. Additionally, Ortiga-Carvalho et al. [69] reported that leptin, which is directly related to body fat mass, acts as a direct stimulator of TSH synthesis. Hence, the effect of ΔBody fat % on TSH reduction in the high TSH group is explained.

Not only do the physical activities of the marine exercise intervention program influence body fat %, but they may also affect the physical ability of individuals. The exercise stress score was calculated using the maximum HR during a series of exercises. Therefore, we figured that physical fitness ability would increase if the exercise stress score decreased. ΔExercise stress through the program correlated with ΔfT4 levels in the high fT4 group. Moreover, ΔExercise stress negatively influenced ΔfT4 after the program. Van den Beld et al. [70] reported that fT4 levels were negatively associated with physical performance in older adult men, which is consistent with our findings. Although the method of obtaining the physical ability score in our study is different, the study is similar in that it obtained physical performance scores to observe the association with fT4. However, Dueñas et al. [71] reported no association between physical activity scores and fT4 levels. We speculate that this difference in results is because these studies focused on the relationship of fT4 levels to the intensity of physical activities and exercise, while our results investigated the relationship of fT4 to changes in physical ability.

Participants were exposed to the natural environment while participating in the program, which is beneficial for psychological restoration and balance of the ANS [31,72,73]. Most studies have shown that natural exposure lowers the activation of the SNS of the ANS and increases the activation of the PNS using HRV [73,74]. Here, no HRV parameter results significantly changed after the retreat program. However, ΔnLF, ΔLF/HF ratio, and ΔTSH levels showed significant positive correlations, especially in the high TSH group. This indicates that the decrease in sympathetic nervous activation is associated with a TSH decrease in the high TSH group, since nLF and LF/HF ratio are indexes of sympathetic activity compared with that of parasympathetic activity [75]. Moreover, nLF was found to be a major factor influencing ΔTSH levels. This result is consistent with many studies that showed a significant association between sympathetic activation and thyroid function, including TSH levels, using HRV. Karthik et al. [76] and Syamsunder et al. [77] reported an association between hypothyroid sympathovagal imbalance due to sympathetic activation by an increased LF/HF ratio. Additionally, Kabir et al. [27] found that the LF and LF/HF ratio showed a significant positive correlation with TSH levels, implying that SNS activation becomes more active as TSH increases. Additionally, Brusseasu et al. [78] recently reported a significant association where the nLF and LF/HF ratio were significantly decreased by the change of hormone levels to the normal range by hyperthyroid treatment. However, all the studies mentioned above focused on participants with thyroid dysfunction, such as hypo- and hyperthyroidism. In our study, we found that sympathetic and parasympathetic changes were also associated with changes in TSH levels, even in euthyroid participants.

This study has several limitations. First, the long-term effects of the marine exercise retreat program should be investigated. We conducted the program for 5 d and confirmed the short-term effects on thyroid-related hormones. However, it is necessary to check whether the program has a long-term effect using a longer program. Second, this study used a relatively small sample size. Further studies are needed with a larger sample size, including the status of the participants. Lastly, iodine level information from participants is needed for further analysis of thyroid changes, since iodine level is known to have a high association with thyroid function.

## 5. Conclusions

This study examined the effects of a marine exercise retreat program on thyroid function by measuring TSH and fT4 levels in middle-aged euthyroid women. Our study shows that the marine exercise retreat program may reduce TSH and fT4 levels, especially in the high group, showing an effective reduction. We investigated the influence of TSH and fT4 level changes after the program. We found that changes in nLF and CO exposure levels affected TSH level changes in the total participants. In the high TSH group, TSH changes were significantly affected by changes in body fat, normalized LF, and NO_2_ exposure. Additionally, the fT4 levels in the total participants were significantly affected by changes in BMI, NO_2_ exposure, and PM_10_ exposure. It was found that the change in BMI and CO exposure in the low fT4 group and the change in exercise stress in the high fT4 group affected fT4 changes.

## Figures and Tables

**Figure 1 ijerph-20-01542-f001:**
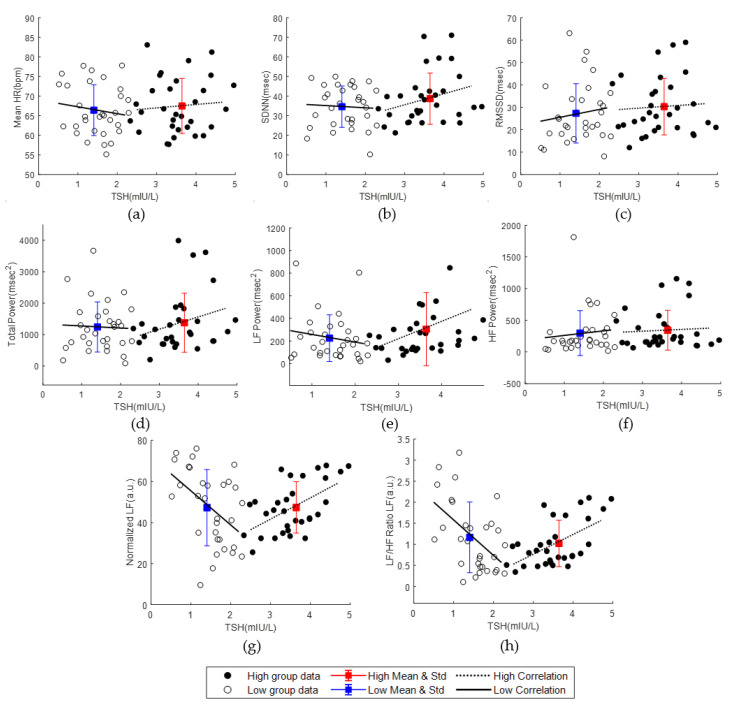
Scatter plot with mean and standard deviation, best fit lines, and correlations between thyroid stimulating hormone (TSH) and heart rate variability (HRV) parameters in the high and low TSH groups before the program (high group: black circle and dotted fit line, low group: blank circle and solid fit line) The correlations (r) and significance (*P*) of the low and high groups are indicated at the bottom left and right of each graph, respectively. (**a**) Mean heart rate (HR); (**b**) standard deviation of NN intervals (SDNN); (**c**) root mean square of successive NN interval differences (RMSSD); (**d**) total power; (**e**) low frequency (LF) power; (**f**) high frequency (HF) power; (**g**) Normalized LF; (**h**) LF/HF Ratio.

**Figure 2 ijerph-20-01542-f002:**
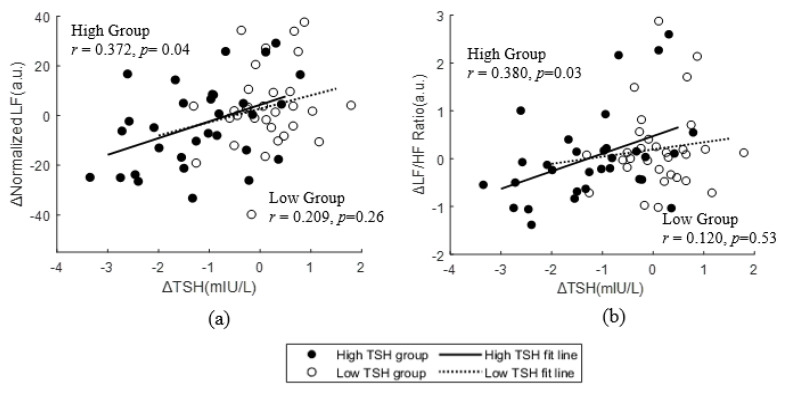
Scatter plots, best-fit lines, and correlations of TSH change and normalized LF change/TSH change and LF/HF ratio changes in the program. (**a**) ΔNormalized LF and ΔTSH of the high and low TSH subgroups, (**b**) ΔLF/HF ratio and ΔTSH of the high and low TSH subgroups.

**Table 1 ijerph-20-01542-t001:** Baseline characteristics, physical body measurement, and thyroid-related hormone of participants and comparisons between the high and low TSH and fT4 groups.

Variable	Total (*n* = 62)	TSH			fT4		
		Low (*n* = 31)	High (*n* = 31)	*p*-Value	Low (*n* = 31)	High (*n* = 31)	*p*-Value
Baseline Characteristics							
Age (years)	58.5 ± 4.8	58.4 ± 5.6	58.6 ± 4.1	0.68	58.2 ± 5.5	58.7 ± 4.1	0.68
Residence							
Seoul	27	12 (%)	15 (%)	0.20	18	9	0.05
Gyeonggi/Gwangju	26	12 (%)	14 (%)		12	14	
Wando	12	9 (%)	3 (%)		3	9	
Hypertension History							
Yes	11	6 (%)	5 (%)	0.93	25	24	0.83
No	54	26 (%)	23 (%)		6	5	
Diabetes History							
Yes	6	3 (%)	3 (%)	0.93	2	4	0.34
No	59	27 (%)	25 (%)		28	24	
Hyperlipidemia History							
Yes	15	6 (%)	9 (%)	0.37	9	6	0.46
No	50	24 (%)	19 (%)		21	22	
Smoking							
Never	61	31 (%)	30 (92.0%)	0.85	30	28	0.61
Past	4	2 (%)	2 (8.0%)		1	3	
Current	0	0 (%)	0 (0.0%)		0	0	
Physical body measurements							
BMI (kg/m^2^)	24.4 ± 2.4	24.6 ± 2.8	24.2 ± 2.0	0.53	24.6 ± 2.7	24.2 ± 2.0	0.53
Body Fat (%)	33.9 ± 5.68	34.1 ± 5.73	33.8 ± 4.89	0.87	34.1 ± 5.7	33.8 ± 4.9	0.87
Exercise stress score (a.u.)	75.3 ± 15.6	73.5 ± 16.6	77.1 ± 14.6	0.36	73.5 ± 16.6	77.1 ± 14.6	0.36
Blood Pressure							
SBP (mmHg)	117 ± 16	117 ± 16	117 ± 15	0.95	117 ± 17	117 ± 16	0.91
DBP (mmHg)	76.3 ± 10	76.2 ± 11	76.2 ± 9.3	0.94	76.3 ± 11	76.2 ± 9.3	0.98
Thyroid-related hormone							
TSH (mIU/L)	2.50 ± 1.2	1.47 ± 0.51	3.56 ± 0.66	<0.01	2.66 ± 1.15	2.32 ± 1.25	0.26
fT4 (ng/dL)	1.24 ± 0.12	1.23 ± 0.14	1.25 ± 0.12	0.66	1.14 ± 0.067	1.35 ± 0.081	<0.01

Abbreviations: BMI, body mass index; SBP, systolic blood pressure; DBP, diastolic blood pressure; TSH, thyroid stimulating hormone; fT4, free thyroxine.

**Table 2 ijerph-20-01542-t002:** Comparison of air pollution concentrations in residential areas of participants in the TSH and fT4 level groups.

Variable	Total (*n* = 62)	TSH			fT4		
		Low (*n* = 31)	High (*n* = 31)	*p*-Value	Low (*n* = 31)	High (*n* = 31)	*p*-Value
SO_2_ (ppm)	0.00304 ± 0.00067	0.00299 ± 0.00075	0.00311 ± 0.00059	0.49	0.00323 ± 0.00069	0.00286 ± 0.00062	0.03
NO_2_ (ppm)	0.0205 ± 0.0088	0.0188 ± 0.0094	0.0222 ± 0.0080	0.12	0.0236 ± 0.0081	0.0175 ± 0.0086	0.01
CO (ppm)	0.502 ± 0.071	0.495 ± 0.061	0.509 ± 0.079	0.44	0.513 ± 0.078	0.491 ± 0.0611	0.21
O_3_ (ppm)	0.0259 ± 0.0038	0.0262 ± 0.0037	0.0256 ± 0.0039	0.53	0.0249 ± 0.0042	0.0269 ± 0.00308	0.04
PM_10_ (μg/m^3^)	35.2 ± 6.4	34.3 ± 7.3	36.2 ± 5.2	0.24	36.9 ± 5.4	33.6 ± 6.9	0.04
PM_2.5_ (μg/m^3^)	19.2 ± 4.0	18.3 ± 4.3	20.1 ± 3.4	0.07	19.9 ± 3.2	18.5 ± 4.6	0.17

Abbreviations: TSH, thyroid stimulating hormone; fT4, free thyroxine.

**Table 3 ijerph-20-01542-t003:** Comparisons of HRV parameters of participants between the high and low TSH and fT4 groups.

Variable	Total (*n* = 62)	TSH			fT4		
		Low (*n* = 31)	High (*n* = 31)	*p*-Value	Low (*n* = 31)	High (*n* = 31)	*p*-Value
Time Domain							
Mean HR (bpm)	66.9 ± 6.7	66.4 ± 6.5	67.5 ± 7.0	0.54	66.2 ± 5.4	67.6 ± 7.8	0.41
SDNN (ms)	36.6 ± 12.0	34.6 ± 10.5	38.7 ± 13.1	0.18	37.5 ± 11.1	35.7 ± 12.8	0.56
RMSSD (ms)	28.7 ± 12.9	27.3 ± 13.3	30.3 ± 12.6	0.37	29.6 ± 12.3	27.9 ± 13.7	0.63
Frequency Domain							
Total power (ms^2^)	1370 ± 990	1240 ± 798	1500 ± 1156	0.31	1380 ± 1048	1360 ± 953	0.94
LF power (ms^2^)	265 ± 272	224 ± 206	304 ± 324	0.25	286 ± 337	243 ± 491	0.54
HF power (ms^2^)	318 ± 332	295 ± 354	342 ± 313	0.58	342 ± 305	296 ± 361	0.59
Normalized LF	47.3 ± 15.7	47.2 ± 18.5	47.4 ± 12.5	0.97	45.3 ± 13.5	49.3 ± 17.6	0.32
LF/HF Ratio	1.09 ± 0.71	1.16 ± 0.84	1.02 ± 0.55	0.43	0.962 ± 0.581	1.22 ± 0.80	0.14

Abbreviations: Mean HR, Mean Heartrate; SDNN, standard deviation of the NN interval; RMSSD, Root Mean Square of the Successive Differences; LF Power, Low Frequency Power; HF Power, high frequency power; Normalized LF, Normalized low frequency; LF/HF Ratio, Low frequency/high frequency ratio.

**Table 4 ijerph-20-01542-t004:** Comparison of physical body measurement and thyroid related hormones of participants before and after the marine exercise retreat program.

Variables		Total (*n* = 62)	TSH		fT4	
			Low (*n* = 31)	High (*n* = 31)	Low (*n* = 31)	High (*n* = 31)
BMI (kg/m^2^)	Before	24.4 ± 2.4	24.6 ± 2.8	24.2 ± 2.0	24.6 ± 2.7	24.2 ± 2.0
	After	24.4 ± 2.4	24.7 ± 2.7	24.1 ± 2.0	24.1 ± 2.4	24.7 ± 2.4
	*p*-value	0.59	0.56	0.03	0.64	0.82
Body Fat% (%)	Before	33.9 ± 5.68	34.1 ± 5.73	33.8 ± 4.89	34.1 ± 5.7	33.8 ± 4.9
	After	33.8 ± 5.1	33.9 ± 5.8	33.8 ± 4.5	33.4 ± 5.4	34.3 ± 4.9
	*p*-value	0.61	0.56	0.89	0.66	0.77
Exercise stress score (a.u.)	Before	75.3 ± 15.6	73.5 ± 16.6	77.1 ± 14.6	73.5 ± 16.6	77.1 ± 14.6
	After	71.9 ± 11.9	69.9 ± 11.4	74.0 ± 12.1	73.1 ± 12.1	70.7 ± 11.6
	*p*-value	0.06	0.12	0.26	0.32	0.10
Blood Pressure						
SBP (mmHg)	Before	117 ± 16	117 ± 16	117 ± 15	117 ± 17	117 ± 16
	After	114 ± 13	115 ± 14	113 ± 11	113 ± 13	116 ± 13
	*p*-value	0.14	0.38	0.23	0.17	0.55
DBP (mmHg)	Before	76.3 ± 10	76.2 ± 11	76.2 ± 9.3	76.3 ± 11	76.2 ± 9.3
	After	75.7 ± 8.6	76.6 ± 9.6	74.8 ± 7.5	74.8 ± 8.0	76.6 ± 9.2
	*p*-value	0.73	0.87	0.54	0.58	0.94
Thyroid						
TSH (mIU/L)	Before	2.50 ± 1.2	1.47 ± 0.51	3.56 ± 0.66	2.66 ± 1.15	2.32 ± 1.25
	After	2.00 ± 0.95	1.62 ± 0.81	2.38 ± 0.94	2.28 ± 1.03	1.70 ± 0.77
	*p*-value	<0.01	0.21	<0.01	0.09	<0.01
fT4 (ng/dL)	Before	1.24 ± 0.12	1.23 ± 0.14	1.25 ± 0.12	1.14 ± 0.067	1.35 ± 0.081
	After	1.17 ± 0.11	1.17 ± 0.13	1.17 ± 0.09	1.12 ± 0.11	1.21 ± 0.10
	*p*-value	<0.01	<0.01	<0.01	0.55	<0.01

SBP, systolic blood pressure; DBP, diastolic blood pressure;TSH, thyroid stimulating hormone; fT4, free thyroxine.

**Table 5 ijerph-20-01542-t005:** Comparison of HRV of participants before and after the marine exercise retreat program.

Variables		Total (*n* = 62)	TSH		fT4	
			Low (*n* = 31)	High (*n* = 31)	Low (*n* = 31)	High (*n* = 31)
Time Domain						
Mean HR (bpm)	Before	66.9 ± 6.7	66.4 ± 6.5	67.5 ± 7.0	66.2 ± 5.4	67.6 ± 7.8
	After	69.4 ± 7.2	68.7 ± 7.3	70.2 ± 7.2	69.4 ± 6.5	69.4 ± 8.0
	*p*-value	<0.01	0.03	<0.01	<0.01	0.02
SDNN (ms)	Before	36.6 ± 12.0	34.6 ± 10.5	38.7 ± 13.1	37.5 ± 11.1	35.7 ± 12.8
	After	36.0 ± 13.0	33.8 ± 11.3	38.2 ± 14.5	35.3 ± 11.4	36.7 ± 15.7
	*p*-value	0.74	0.74	0.88	0.42	0.73
RMSSD (ms)	Before	28.7 ± 12.9	27.3 ± 13.3	30.3 ± 12.6	29.6 ± 12.3	27.9 ± 13.7
	After	27.4 ± 12.4	25.0 ± 12.5	29.9 ± 12.1	25.2 ± 10.2	29.6 ± 14.2
	*p*-value	0.46	0.42	0.88	0.08	0.53
Frequency domain						
TP (ms^2^)	Before	1370 ± 990	1240 ± 798	1500 ± 1156	1380 ± 1048	1360 ± 953
	After	1170 ± 1030	1030 ± 1060	1288 ± 1225	1020 ± 530	1320 ± 1360
	*p*-value	0.23	0.29	0.46	0.08	0.88
LF (ms^2^)	Before	265 ± 272	224 ± 206	304 ± 324	286 ± 337	359 ± 499
	After	240 ± 274	211 ± 198	269 ± 335	202 ± 154	278 ± 355
	*p*-value	0.58	0.64	0.67	0.20	0.57
HF (ms^2^)	Before	318 ± 332	295 ± 354	342 ± 313	342 ± 305	299 ± 363
	After	284 ± 260	231 ± 257	337 ± 257	238 ± 217	330 ± 293
	*p*-value	0.43	0.33	0.94	0.05	0.38
Normalized LF	Before	47.3 ± 15.7	47.2 ± 18.5	47.4 ± 12.5	45.3 ± 13.5	49.3 ± 17.6
	After	47.2 ± 18.9	50.7 ± 19.0	43.7 ± 18.5	48.9 ± 19.0	45.5 ± 18.9
	*p*-value	0.97	0.25	0.24	0.25	0.21
LF/HF Ratio	Before	1.09 ± 0.71	1.16 ± 0.84	1.02 ± 0.55	0.962 ± 0.581	1.22 ± 0.80
	After	1.22 ± 0.99	1.38 ± 1.01	1.06 ± 0.95	1.30 ± 1.01	1.13 ± 0.97
	*p*-value	0.29	0.18	0.85	0.05	0.53

Mean HR, Mean Heartrate; SDNN, standard deviation of the NN interval; RMSSD, Root Mean Square of the Successive Differences; LF Power, Low Frequency Power; HF Power, high frequency power; Normalized LF, Normalized low frequency; LF/HF Ratio, Low frequency/high frequency ratio.

**Table 6 ijerph-20-01542-t006:** Comparison of exposed air pollutant concentration before and after the marine exercise retreat program.

Variables		Total	TSH		fT4	
			Low	High	Low	High
SO_2_ (ppm)	Before	0.00304 ± 0.00067	0.00299 ± 0.00075	0.00311 ± 0.00059	0.00323 ± 0.00069	0.00286 ± 0.00062
	After	0.00276 ± 0.00030	0.00278 ± 0.00030	0.00274 ± 0.00030	0.00269 ± 0.00032	0.00283 ± 0.00026
	*p*-value	<0.01	0.16	<0.01	<0.01	0.79
NO_2_ (ppm)	Before	0.0205 ± 0.0088	0.0188 ± 0.0094	0.0222 ± 0.0080	0.0236 ± 0.0081	0.0175 ± 0.0086
	After	0.00511 ± 0.00130	0.00494 ± 0.0013	0.00530 ± 0.0012	0.00533 ± 0.0014	0.00123 ± 0.0012
	*p*-value	<0.01	<0.01	<0.01	<0.01	<0.01
CO (ppm)	Before	0.502 ± 0.071	0.495 ± 0.061	0.509 ± 0.079	0.513 ± 0.078	0.491 ± 0.0611
	After	0.462 ± 0.065	0.468 ± 0.057	0.454 ± 0.057	0.462 ± 0.064	0.461 ± 0.067
	*p*-value	<0.01	0.06	<0.01	<0.01	0.06
O_3_ (ppm)	Before	0.0259 ± 0.0038	0.0262 ± 0.0037	0.0256 ± 0.0039	0.0249 ± 0.0042	0.0269 ± 0.00308
	After	0.0326 ± 0.0062	0.0327 ± 0.0058	0.0325 ± 0.032	0.0321 ± 0.0066	0.0331 ± 0.0058
	*p*-value	<0.01	<0.01	<0.01	<0.01	<0.01
PM_10_ (μg/m^3^)	Before	35.2 ± 6.4	34.3 ± 7.3	36.2 ± 5.2	36.9 ± 5.4	33.6 ± 6.9
	After	29.1 ± 9.8	27.4 ± 9.6	30.8 ± 9.8	29.7 ± 9.7	28.3 ± 9.9
	*p*-value	<0.01	<0.01	<0.01	<0.01	<0.01
PM_2.5_ (μg/m^3^)	Before	19.2 ± 4.0	18.3 ± 4.3	20.1 ± 3.4	19.9 ± 3.2	18.5 ± 4.6
	After	11.7 ± 1.7	11.5 ± 1.6	11.8 ± 1.9	11.6 ± 1.8	11.6 ± 1.7
	*p*-value	<0.01	<0.01	<0.01	<0.01	<0.01

**Table 7 ijerph-20-01542-t007:** Multiple regression analysis of factors influencing TSH changes in the marine exercise program.

Thyroid Hormone	Group	Variables	B	*β*	SE	T	*p*
ΔTSH	Total ^(1)^						
		ΔNormalized LF (a.u.)	0.0264	0.348	0.008	3.116	<0.01
		ΔCO (ppm)	4.24	0.362	1.309	3.236	<0.01
	High ^(2)^						
		ΔBody fat% (%)	0.237	0.245	0.108	2.20	0.04
		ΔNormalized LF (a.u.)	0.025	0.268	0.011	2.35	0.03
		ΔNO_2_ (ppm)	58.95	0.674	9.95	5.92	<0.01

(1) Total: adjusted R^2^ = 0.240, *p*-value < 0.01; (2) High: adjusted R^2^ = 0.621, *p*-value < 0.01. Abbreviations: ΔTSH: changes of TSH, ΔCO: changes of CO, ΔBody fat %: changes of body fat %, ΔNO_2_: changes of NO_2_.

**Table 8 ijerph-20-01542-t008:** Multiple regression analysis of factors influencing fT4 changes in the marine exercise program.

Thyroid Hormone	Group	Variables	B	*β*	SE	t	*p*
ΔfT4	Total ^(1)^						
		ΔBMI (kg/m^2^)	−0.111	−0.328	0.038	−2.94	<0.01
		ΔNO_2_ (ppm)	−3.64	−0.261	1.557	−2.34	0.02
		ΔPM_10_ (ppm)	−0.003	−0.254	0.001	−2.27	0.03
	Low ^(2)^						
		ΔBMI	−0.139	−0.455	0.042	−3.30	<0.01
		ΔCO (ppm)	−0.495	−0.463	0.015	−3.36	<0.01
	High ^(3)^						
		ΔExercise Stress (a.u.)	−0.0023	−0.407	0.001	−2.36	0.02

(1) Total: adjusted R^2^ = 0.296, *p*-value < 0.01; (2) Low: adjusted R^2^ = 0.440, *p*-value < 0.01; (3) High: adjusted R^2^ = 0.136, *p*-value = 0.02. Abbreviations: ΔfT4, changes of fT4; ΔBMI: changes of BMI; ΔPM_10_, changes of PM_10_; ΔNO_2_, changes of NO_2_; ΔCO, changes of CO; ΔExercise Stress: changes of exercise stress score.

## Data Availability

The data that support the findings of this study are available on request from the corresponding author. The data are not publicly available due to privacy of research participants.

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
