# Peer review of "Effect of the Marine Exercise Retreat Program on Thyroid-Related Hormones in Middle-Aged Euthyroid Women"

_ijerph, 2023, doi:10.3390/ijerph20021542_

Round 1
Reviewer 1 Report
Interesting study on the interaction of the marine environment with human health. This is an original scientific research in the field of "Blue Health", much praised by the media and popular texts.
"Blue health" is well-being linked to an environment, particularly healthy, and rich in beneficial elements: iodine, selenium for the proper functioning of the thyroid, together with low concentrations of heavy metals and particulates (PM 10, PM 2,5…) ; ultraviolet rays for the production of VIt D, to fight osteopenia and osteoporosis and improve the immune system; walking on dry sand or in water to promote venous and lymphatic return; until it has positive effects on nutrition, breathing and mood.
The authors have chosen an exceptional setting for their study: the splendid Sinji island in Wando-gun, pearl of the southern coast and first blue flag of Korea. The environment is particularly healthy and promotes positive lifestyles.
The study is well designed and well conducted.
It would be interesting to know why only a female population was chosen to study.
The introduction is exhaustive and interesting but could benefit from a brief note on the generic benefits of the marine environment.
Author Response
Dear reviewer,
I appreciate the time and effort to providing valuable feedback on my manuscript. I am grateful to receive your insight on my paper.
Please see the attachment.

Reviewer 2 Report
1. In this study, the authors analyzed the influence o marine exercise on thyroid function. The manuscript is well written, and the results are novel. However, some variables were uncontrolled in this study. For example, the influence of altitude and climate on thyroid function. In this way, urban, suburban, and rural areas should be at the same altitude level and similar environmental temperatures. Moreover, four exercise sessions is a little time to modify hormone levels, which is the explanation for this finding. Perhaps, it is necessary to have a group with similar environmental conditions, except exercise to confirm the effect of marine activity on thyroid profile. Other observations are:
1. Although the importance of gases was described in the third paragraph of the introduction, it was not put in the research question.
2. As a recommendation, abbreviations should be described in the legend of all tables.
3. In my opinion, a logistic regression analysis could be more appropriate to include all variables and not have a separate analysis. In this way, the residential pollutant exposure should be analyzed according to urban, suburban, and rural areas, including this in the analysis together with the intervention regimen.
4. The conclusion of this study is difficult to be tested because the statistical analysis does not integrate all variables.
Author Response

(The authors gave the same response as above.)
